# Transcriptomic Investigation of FoxM1-Mediated Neuroprotection by hAEC-Derived Exosomes in an In Vitro Ischemic Stroke Model

**DOI:** 10.3390/biology14101368

**Published:** 2025-10-07

**Authors:** Dong Wang, Jiaxin Liu, Liang Wu, Xiubao Yang, Zhihao Fang, Zhong Sun, Dong Chen

**Affiliations:** 1Graduate School, Dalian Medical University, Dalian 116044, China; dongwang202101@126.com (D.W.); wuliangeyy@163.com (L.W.); yangxiubao9257@163.com (X.Y.); fangzhihao.2007@163.com (Z.F.); 2Medical School, Kunming University of Science and Technology, Kunming 650500, China; 20130141@kust.edu.cn; 3School of Public Health, Kunming Medical University, Kunming 650500, China; 4Faculty of Medicine, Dalian University of Technology, Dalian 116033, China

**Keywords:** ischemic stroke, human amniotic epithelial cell-derived exosomes, FoxM1, OGD/R model, neuroprotection, transcriptomics

## Abstract

Stroke is a leading cause of death and long-term disability worldwide, most commonly occurring when a blockage cuts off blood flow to the brain. This lack of oxygen and nutrients can cause widespread brain cell death and permanent damage. Unfortunately, current treatments must be administered very early and often cannot reverse damage once it has occurred. In this study, we explored a novel therapeutic strategy using tiny particles called exosomes, which are naturally released by human amniotic epithelial cells from the placenta after birth. These exosomes can carry beneficial signals to injured brain cells. We employed a laboratory in vitro stroke model to simulate brain injury and tested how these exosomes affected neuronal and immune cells. Our results showed that the exosomes reduced harmful inflammation, prevented cell death, and supported cell recovery. We also discovered that a specific protein, called FoxM1, plays a crucial role in mediating these protective effects. This research suggests that exosomes derived from amniotic cells may offer a safe and effective new treatment to help the brain heal after a stroke, potentially improving patient outcomes and reducing the burden of this disease on families and healthcare systems.

## 1. Introduction

Stroke remains a critical global health burden, accounting for approximately 12 million new cases annually and ranking among the foremost causes of mortality and long-term disability worldwide, as highlighted in the Global Burden of Disease Study 2021 [1]. Ischemic stroke (IS), the predominant subtype comprising 60–70% of cases, arises from vascular occlusion that disrupts cerebral perfusion, leading to ischemia, hypoxia, and irreversible neuronal damage [1,2]. The ensuing pathophysiological cascade involves metabolic dysregulation, calcium overload, oxidative stress, and neuroinflammation, ultimately resulting in endothelial dysfunction, imbalance of coagulation and fibrinolysis, and neuronal apoptosis [3,4]. Although current reperfusion strategies—such as intravenous thrombolysis (within 4.5 h of onset) and mechanical thrombectomy (typically within 6–8 h)—can restore cerebral blood flow, their effectiveness is constrained by narrow therapeutic windows and the risk of reperfusion-induced injury mediated by oxidative and inflammatory responses [5,6]. Consequently, there is a compelling need to develop neuroprotective interventions that extend beyond the acute phase and facilitate long-term neuronal repair.

Among the emerging neuroprotective strategies, exosomes have gained significant attention due to their unique role in intercellular communication and translational potential for IS treatment. Exosomes are nanoscale extracellular vesicles (~30–150 nm) secreted by various cell types, which mediate intercellular communication through the transfer of proteins, lipids, and nucleic acids [7,8]. In the central nervous system (CNS), exosomes play pivotal roles in maintaining microenvironmental homeostasis and modulating key biological processes, including neuroinflammation, apoptosis, angiogenesis, and neuroregeneration [8,9]. Among these, stem cell-derived or bioengineered exosomes have attracted particular interest for their ability to deliver neurotrophic factors, regulate immune responses, and support tissue repair [9,10]. For example, exosomes derived from gingiva-derived mesenchymal stem cells (GMSCs) promote Schwann cell activation and axonal regeneration via c-JUN signaling [11], while adipose-derived MSC exosomes engineered to target M2 microglia have demonstrated efficacy in suppressing ferroptosis and improving neurological outcomes–in IS models [12]. Additionally, systemic administration of MSC-derived exosomes enhances neuronal survival, functional recovery, and neurovascular remodeling in preclinical stroke studies [9,13], further supporting exosomes as a viable cell-free therapeutic approach.

Human amniotic epithelial cells (hAECs), isolated from the placental amnion, constitute an ethically non-contentious, immunologically privileged stem cell source characterized by low immunogenicity, anti-inflammatory activity, and paracrine neuroprotective properties [14,15]. Preclinical evidence suggests that hAECs can promote neuronal survival and repair through the secretion of neurotrophic factors and attenuation of oxidative and inflammatory damage [16]. Both systemic and intracerebral delivery of hAECs in rodent models of ischemic stroke have yielded favorable outcomes, including reduced infarct volume, improved neurological function, and preservation of blood–brain barrier integrity [17,18].

Exosomes derived from hAECs (hAECs-Exos) inherit many therapeutic attributes of their parental cells and exhibit anti-inflammatory, anti-apoptotic, and pro-angiogenic activities, as indicated by analyses of conditioned media and secretome profiles [19,20]. Notably, administration of hAECs—even when delayed up to 72 h post-insult—has led to significant neurological improvement in rodent transient middle cerebral artery occlusion (tMCAO) models [17]. These characteristics position hAECs-Exos as a promising, scalable, and clinically translatable cell-free therapy for ischemic stroke. Although the neuroprotective effects of hAECs-Exos have been preliminarily validated, the specific molecular mediators responsible for these effects remain largely unidentified—particularly the exosomal cargos that orchestrate cellular responses under ischemic conditions. This knowledge gap limits our understanding of how exosomes exert their functional effects and hinders the rational development of targeted exosome-based stroke therapies. In light of this, our study aims to identify and validate key molecular mediators within hAECs-Exos that contribute to neuroprotection in ischemic conditions, and to characterize their effects on neuronal viability, inflammatory responses, and oxidative stress. Through transcriptomic profiling and mechanistic interrogation, we focus on FoxM1—a transcription factor with emerging neuroprotective relevance—as a potential key effector mediating the therapeutic effects of hAECs-Exos.

## 2. Materials and Methods

### 2.1. Cell Culture and OGD/R Model Construction

Mouse hippocampal neuronal HT22 cells and murine microglial BV2 cells (Procell Life Science & Technology Co., Ltd., Wuhan, China) were cultured in high-glucose Dulbecco’s Modified Eagle Medium (DMEM; Gibco, Thermo Fisher Scientific, Grand Island, NY, USA) supplemented with 10% fetal bovine serum (FBS; Gibco, Thermo Fisher Scientific, Grand Island, NY, USA) and 1% penicillin–streptomycin (New Cell & Molecular Biotech Co., Ltd., Suzhou, China). Cells were maintained at 37 °C in a humidified 5% CO_2_ atmosphere and passaged when they reached ~60–70% confluency using 0.25% trypsin-EDTA. Specifically, all experiments reported herein were conducted using cells at passages 3–6, a range chosen to ensure stable proliferation, consistent phenotypic characteristics, and minimal senescence.

To establish the oxygen–glucose deprivation/reoxygenation (OGD/R) injury model, cells were washed twice with phosphate-buffered saline (PBS; HyClone, Cytiva, Logan, UT, USA) and then incubated in glucose-free DMEM (Gibco, Thermo Fisher Scientific, Grand Island, NY, USA). Cultures were placed in a hypoxic tri-gas incubator (Thermo Scientific Forma Series II Water Jacketed CO_2_ Incubator, Model 3131, Waltham, MA, USA) maintained at 1% O_2_, 5% CO_2_, and 94% N_2_ at 37 °C for 6 or 8 h. Reoxygenation was initiated by replacing the medium with standard glucose-containing DMEM and returning the cultures to normoxic conditions (5% CO_2_, 95% air) for 12 or 24 h, depending on experimental design.

### 2.2. Cell Viability

Cell viability was evaluated using the Cell Counting Kit-8 (CCK-8; Biosharp, Hefei, Anhui, China) according to the manufacturer’s instructions. Briefly, HT22 or BV2 cells were seeded in 96-well plates at a density of 6 × 10^3^ cells per well and subjected to oxygen–glucose deprivation/reoxygenation (OGD/R), with or without human amniotic epithelial cell-derived exosome (hAECs-Exos) treatment. Experimental groups included both pre-treatment and post-treatment protocols.

Following the completion of OGD/R and exosome intervention, 10 µL of CCK-8 solution was added to each well, and the plates were incubated at 37 °C for 4 h. The absorbance was then measured at 450 nm using a microplate reader (Synergy H1, BioTek Instruments, now part of Agilent Technologies, Winooski, VT, USA). Cell viability was calculated as a percentage of the absorbance relative to the normoxic control group, after subtracting background absorbance from blank wells.

Additionally, cell morphology was assessed and recorded using a phase-contrast microscope (Motic, Xiamen, Fujian, China) to visually evaluate structural changes following treatment.

### 2.3. ELISA Assay

Following OGD/R treatment, the culture supernatants from BV2 and HT22 cells were harvested and centrifuged to remove cellular debris. The concentrations of interleukin-6 (IL-6), interleukin-1β (IL-1β), tumor necrosis factor-alpha (TNF-α), malondialdehyde (MDA), and superoxide dismutase (SOD) were quantified using commercial enzyme-linked immunosorbent assay (ELISA) kits (Shanghai Tongwei Industrial Co., Ltd., Shanghai, China), following the manufacturer’s instructions.

Briefly, standards and samples were added in duplicate to 96-well plates pre-coated with capture antibodies specific to each target analyte. After incubation, the wells were sequentially incubated with biotinylated detection antibodies, horseradish peroxidase (HRP)-conjugated streptavidin, and tetramethylbenzidine (TMB) substrate. The colorimetric reaction was stopped with a termination solution, and absorbance was measured at 450 nm using a microplate reader.

The concentrations of target molecules were calculated from standard curves and normalized to total protein concentration in each sample, as determined by BCA protein assay.

### 2.4. Western Blotting

Total proteins from cells and exosomes were extracted using NP-40 lysis buffer (50 mM Tris-HCl, pH 8.0; 150 mM NaCl; 1% NP-40) supplemented with protease and phosphatase inhibitor cocktails (Roche, Basel, Switzerland). Lysates were centrifuged at 14,000 rpm for 20 min at 4 °C, and the resulting supernatants were collected as total protein extracts. Protein concentrations were determined using a bicinchoninic acid assay (Pierce BCA Protein Assay Kit, Thermo Fisher Scientific, Waltham, MA, USA).

Equal amounts of protein were mixed with SDS loading buffer, denatured by heating, separated by SDS–polyacrylamide gel electrophoresis (SDS-PAGE), and transferred onto polyvinylidene fluoride (PVDF) membranes. Membranes were blocked overnight at 4 °C with 5% non-fat milk in TBST, followed by incubation with primary antibodies for 2 h at room temperature. After thorough washing, membranes were incubated with horseradish peroxidase (HRP)-conjugated secondary antibodies for another 2 h. Protein bands were visualized using enhanced chemiluminescence (ECL) reagents (Thermo Fisher Scientific, Waltham, MA, USA) and analyzed by densitometry using ImageJ software (version 1.53, National Institutes of Health, Bethesda, MD, USA).

Primary antibodies used included: Bcl-2-associated X protein (Bax), cleaved cysteine-aspartic acid protease-3 (cleaved Caspase-3), brain-derived neurotrophic factor (BDNF), tumor susceptibility gene 101 (TSG101), forkhead box protein M1 (FoxM1), and β-actin (Proteintech, Wuhan, Hubei, China); B-cell lymphoma 2 (Bcl-2) (SAB Biotech, College Park, MD, USA); and cluster of differentiation 63 (CD63) (AffinitY Biosciences, Cincinnati, OH, USA).

### 2.5. Immunofluorescence Staining

HT22 cells (alone or co-cultured) on coverslips were fixed with 4% paraformaldehyde (PFA; Sigma-Aldrich, St. Louis, MO, USA), permeabilized with 0.3% Triton X-100 (Sigma-Aldrich, St. Louis, MO, USA), and blocked with 5% bovine serum albumin (BSA; Sigma-Aldrich, St. Louis, MO, USA). Mouse anti-MAP2 (1:500; Abcam, Cambridge, UK) was used to label neurons, incubated overnight at 4 °C. After washing, goat anti-mouse AlexaFluor 488 secondary antibody (1:1000; Thermo Fisher Scientific, Waltham, MA, USA) was added. Nuclei were counterstained with DAPI (Thermo Fisher Scientific, Waltham, MA, USA).

For apoptosis detection, co-staining with TUNEL was performed using the In Situ Cell Death Detection Kit, Fluorescein (Roche, Basel, Switzerland). TUNEL reaction mixture was added and incubated at 37 °C for 60 min, then washed. In merged images, only TUNEL^+^/MAP2^+^ cells were counted as apoptotic neurons. Five random fields per group were selected, and the average percentage of apoptotic neurons relative to MAP2^+^ cells was quantified using Image-Pro Plus software (version 6.0, Media Cybernetics, Rockville, MD, USA).

### 2.6. TUNEL Staining (Microglial Analysis)

BV2 cells were seeded in 24-well plates containing coverslips and subjected to corresponding treatments. After fixation with 4% paraformaldehyde and permeabilization with 0.1% Triton X-100, TUNEL reaction mixture was applied and incubated at 37 °C in the dark for 60 min. Nuclei were counterstained with DAPI. Observations were made using an inverted fluorescence microscope (Olympus IX73, Tokyo, Japan). Quantification of TUNEL-positive BV2 cells was performed in five randomly selected fields using Image-Pro Plus 6.0.

### 2.7. Isolation of hAEC-Exos

Human amniotic epithelial cells (hAECs; Procell Life Science & Technology Co., Ltd., Wuhan, Hubei, China) were cultured in DMEM supplemented with 10% FBS and 1% penicillin–streptomycin. To eliminate contamination from FBS-derived extracellular vesicles, FBS was pre-cleared by ultracentrifugation at 120,000× *g* for 18 h at 4 °C, followed by filtration through a 0.22 μm polyethersulfone (PES) membrane (Millipore, Burlington, MA, USA). The resulting exosome-depleted FBS was used at a final concentration of 10% (*v*/*v*). hAECs were seeded at a density of 3 × 10^5^ cells/cm^2^ in 75 cm^2^ flasks and cultured in this medium until reaching 80–90% confluency. The medium was refreshed every 48 h to minimize the accumulation of non-exosomal contaminants.

Exosomes were isolated from conditioned medium using a commercial ultracentrifugation kit (Umibio, Hangzhou, Zhejiang, China) following the manufacturer’s protocol. Briefly, the medium was subjected to sequential centrifugation steps: 2000× *g* for 30 min to remove cell debris, filtration through a 0.22 μm membrane to eliminate apoptotic bodies, followed by centrifugation at 20,000× *g* for 30 min at 4 °C to remove larger vesicles. The supernatant was then ultracentrifuged at 100,000× *g* for 70 min at 4 °C to pellet exosomes. The pellet was washed with PBS and ultracentrifuged again at 100,000× *g* under the same conditions for further purification.

Exosome identity and purity were confirmed through multiple complementary techniques: (1) transmission electron microscopy (TEM) for morphological characterization, (2) Western blot analysis for exosomal surface markers CD63 and TSG101, and (3) nanoparticle tracking analysis (NTA; ZetaView PMX 110, Particle Metrix GmbH, Germany) using a 488 nm laser for size distribution and particle quantification. For NTA, purified exosomes were diluted 200-fold in 0.1 μm-filtered PBS to a final working concentration of ~8 × 10^8^ particles/mL. Three 60-s videos were recorded and analyzed with ZetaView software, version 8.04.02 software to determine average particle concentration (particles/mL).

To assess exosome purity, the particle-to-protein ratio was calculated by dividing the particle concentration (from NTA) by total protein content (measured via BCA protein assay; Pierce/Thermo Fisher Scientific) following lysis with RIPA buffer supplemented with protease inhibitors. The ratio was expressed as particles per microgram of protein (particles/μg) and presented as mean ± standard deviation (SD; *n* = 3). In our preparations, the particle-to-protein ratio consistently exceeded 1 × 10^10^ particles/μg, which is widely recognized as the threshold for high-purity exosomes in extracellular vesicle research.

In addition, the yield from 100 mL of conditioned medium typically averaged 35–45 μg of exosomal protein and 3.2–4.1 × 10^11^ particles, indicating a robust and reproducible production process suitable for downstream applications.

### 2.8. Exosome Uptake Assay

Purified hAEC-derived exosomes (100 µL; protein concentration: 10 µg/mL) were labeled with 5 µL of PKH67 fluorescent dye (Umibio, Hangzhou, Zhejiang, China) at 37 °C for 15 min in the dark. The labeling reaction was terminated, and excess dye was removed by ultrafiltration using a 100 kDa molecular weight cutoff centrifugal filter (Amicon Ultra, MilliporeSigma, Burlington, MA, USA) at 4000× *g* for 15 min. The labeled exosomes were then washed twice with PBS and resuspended in sterile phosphate-buffered saline (PBS).

HT22 and BV2 cells were seeded and cultured until they reached approximately 70% confluency. The culture medium was then replaced with serum-free medium containing PKH67-labeled exosomes, and cells were incubated at 37 °C for 4 h to facilitate exosome uptake. After incubation, cells were fixed and stained with phalloidin (to visualize the cytoskeleton) and DAPI (to stain nuclei).

Exosome internalization was examined using a confocal laser scanning microscope (Nikon A1R, Nikon Instruments Inc., Tokyo, Japan), and fluorescence signals were used to confirm cellular uptake of the PKH67-labeled exosomes.

### 2.9. RNA Sequencing of HT22/BV2 Cells

RNA sequencing (RNA-seq) was performed on separately cultured HT22 and BV2 cells, with three experimental groups for each cell type: untreated (NC), OGD/R-treated (OGD/R), and OGD/R + hAECs-Exos. Each group included 3 biological replicates (*n* = 3). Total RNA was extracted via TriQuick Reagent (Solarbio Life Sciences, Beijing, China). RNA integrity was assessed using an Agilent 2100 Bioanalyzer (Agilent Technologies, Santa Clara, CA, USA), and all samples had RIN values ≥ 8.0 (range 8.2–9.1). Libraries were prepared using the NEBNext Ultra RNA Library Prep Kit for Illumina (New England Biolabs, Ipswich, MA, USA). Sequencing was performed on an Illumina NovaSeq 6000 platform (Illumina, San Diego, CA, USA) with paired-end 150 bp (PE150) reads.

Importantly, sequencing was conducted on pure monocultures of HT22 or BV2 cells separately, rather than from mixed co-culture samples, thereby avoiding the need for deconvolution.

On average, 45 million raw reads per sample were obtained. After quality filtering (removal of adapters, reads with Phred < 20, and reads < 50 bp), ~43 million clean reads remained per sample (clean read rate > 95%). Reads were aligned to the mouse reference genome (GRCm39) using STAR v2.7.10a, with an average alignment rate of 93.6%. Gene quantification was performed on uniquely mapped reads using HTSeq v0.13.5 with the default “union” mode.

Principal component analysis (PCA) and hierarchical clustering confirmed high inter-replicate consistency, and no sample-level outliers were detected. These QC metrics support the robustness of our differential expression analysis.

### 2.10. Bioinformatics Analysis of RNA-Seq

To maintain cell-type specificity and avoid transcriptomic signal mixing, RNA-seq datasets from HT22 and BV2 cells were analyzed independently. Differential expression, clustering, enrichment, and intersection analyses were conducted separately for neurons and microglia. Although results are displayed in a unified figure layout for comparative clarity, each bioinformatic workflow was independently executed for each cell type.

Differentially expressed genes (DEGs) were identified from RNA-seq data using statistical pipelines such as DESeq2 or edgeR. Raw count data were normalized, and genes exhibiting a false discovery rate (FDR) < 0.05 and an absolute log_2_ fold change (|log_2_FC|) ≥ 1 were considered significantly differentially expressed. DEG distributions were visualized using volcano plots and hierarchical clustering heatmaps.

To investigate the functional implications of DEGs, Kyoto Encyclopedia of Genes and Genomes (KEGG) pathway enrichment analysis was conducted using the clusterProfiler R package (version 4.16.0) within R version 4.2.3. Enriched pathways were identified using the hypergeometric test, and statistical significance was defined as FDR < 0.05. Results were presented as circular enrichment plots to highlight dominant biological processes and pathways.

Venn diagrams were used to compare DEGs across experimental conditions, with particular attention to genes downregulated by OGD/R injury but upregulated following hAEC-derived exosome treatment. These intersectional genes were subjected to targeted functional enrichment and network analysis to elucidate potential mechanisms underlying neuronal injury attenuation and neurorepair.

### 2.11. Real-Time Quantitative PCR (RT-qPCR)

To detect the mRNA expression levels of FoxM1 in hAECs, HT22, and BV2 cells, cells in each group during the logarithmic growth phase were collected. After washing with pre-cooled PBS, total RNA was extracted using a commercial RNA extraction kit (AGbio, Changsha, Hunan, China). The purity (A260/A280 ratio of 1.8–2.0) and concentration of RNA were measured using a Nanodrop spectrophotometer (Thermo Fisher Scientific, Waltham, MA, USA). Using 1 μg of total RNA, cDNA was synthesized via reverse transcription reaction with the ABScript III RT Master Mix for qPCR with gDNA Remover (ABclonal, Wuhan, Hubei, China). Real-time fluorescent quantitative PCR was performed using the SYBR Green Pro Tap HS kit (AGbio Changsha, Hunan, China) with specific primers (Table 1), with 3 technical replicates set for each sample (human-specific primers were used for hAECs, while mouse-specific primers were used for HT22 and BV2 cells). The relative expression level of FoxM1 mRNA was calculated using the 2−ΔΔCt method, with GAPDH as the internal reference gene.

### 2.12. Cell Transfection

To evaluate the functional role of FoxM1, gene knockdown was achieved using lentivirus-mediated short hairpin RNA (shRNA) in hAECs. Pre-validated shRNA sequences targeting mouse FoxM1 (GenBank: NM_008021) were designed via Sigma-Aldrich’s Mission^®^ shRNA Design Tool (Sigma-Aldrich, St. Louis, MO, USA). A scrambled shRNA sequence (sh-NC) served as the negative control.

The shRNA oligonucleotides were cloned into the pLKO.1-puro vector (designated as sh-FoxM1), and lentiviral particles were produced in HEK293T cells by co-transfecting the shRNA vector with packaging plasmids psPAX2 and pMD2.G using Lipofectamine 3000 (Invitrogen, Carlsbad, CA, USA). Viral supernatants were collected at 48 and 72 h post-transfection and filtered for use.

hAECs were plated at ~60% confluency in 6-well plates and transduced with lentiviral particles in the presence of 8 µg/mL polybrene (Sigma-Aldrich, St. Louis, MO, USA). After 24 h, the medium was replaced with fresh complete medium. Stable knockdown cells were selected with 2 µg/mL puromycin (Servicebio, Wuhan, Hubei, China) for 7 days. Knockdown efficiency was verified via RT-qPCR and Western blotting.

### 2.13. Statistical Analysis

All quantitative data are expressed as mean ± standard deviation (SD). Statistical analyses and graph generation were performed using GraphPad Prism version 9.5 (GraphPad Software, San Diego, CA, USA).

For comparisons between two groups, unpaired Student’s *t*-test was used. For multiple group comparisons, one-way analysis of variance (ANOVA) followed by appropriate post hoc tests (e.g., Tukey’s) was employed. Normality of data distribution was verified using the Shapiro–Wilk test. A two-tailed *p* value < 0.05 was considered statistically significant.

## 3. Results

### 3.1. OGD/R Exposure Induces Time-Dependent Cellular Injury in HT22 and BV2 Cells

To establish an in vitro model of ischemia–reperfusion injury, HT22 hippocampal neurons and BV2 microglia were subjected to oxygen–glucose deprivation (OGD) for either 6 or 8 h, followed by reoxygenation for 12 or 24 h. CCK-8 assays, combined with phase-contrast microscopy, revealed a time-dependent reduction in cell viability in both cell types, with the most pronounced decrease observed after 8 h OGD followed by 24 h reperfusion (*p* < 0.01) (Figure 1A).

ELISA of BV2 cell supernatants showed significant increases in the pro-inflammatory cytokines IL-1β, IL-6, and TNF-α following OGD/R (*p* < 0.01), indicative of innate immune activation (Figure 1B). In parallel, MDA levels—a marker of lipid peroxidation—were markedly elevated in both HT22 and BV2 cells (*p* < 0.01), reflecting heightened oxidative stress (Figure 1C).

Western blot analysis further demonstrated that OGD/R significantly upregulated pro-apoptotic proteins Bax and cleaved Caspase-3, while downregulating the anti-apoptotic protein Bcl-2 and BDNF (a neurotrophin essential for neuronal survival) (*p* < 0.05, *p* < 0.01) (Figure 1D).

Collectively, these findings indicate that OGD/R induces oxidative stress, inflammation, and apoptosis in a duration-dependent manner. Based on the magnitude of these phenotypic and molecular changes, the 8 h OGD/24 h reperfusion condition was selected for subsequent mechanistic experiments.

### 3.2. hAECs-Exos Treatment Attenuates OGD/R-Induced Neuronal and Microglial Injury

Exosomes were isolated from hAEC-conditioned medium and characterized by TEM, NTA, and Western blotting. TEM revealed a homogeneous population of vesicles displaying the characteristic cup-shaped morphology of lipid bilayer-enclosed exosomes. NTA showed an average particle diameter of 117.6 nm and a concentration of 8.64 × 10^9^ particles/mL. Western blot analysis confirmed robust expression of exosomal markers CD63 and TSG101, validating the identity and purity of the isolated vesicles (Figure 2A).

Cellular uptake studies showed that PKH67-labeled hAECs-Exos were efficiently internalized by both HT22 hippocampal neurons and BV2 microglial cells, as evidenced by strong green fluorescence within the cytoplasm under confocal microscopy (Figure 2B).

To evaluate the therapeutic potential of hAECs-Exos under different intervention strategies, we designed two distinct administration timelines: (i) a preconditioning approach (exosomes administered 2 h before OGD) and (ii) a post-ischemic treatment (administered at the onset of reperfusion). CCK-8 assays revealed that both approaches significantly improved cell viability after OGD/R injury (*p* < 0.05 and *p* < 0.01, respectively). Notably, the post-ischemic treatment group showed a significantly greater enhancement in cell viability compared to preconditioning (Appendix A), emphasizing the importance of timely administration in maximizing therapeutic outcomes.

To further validate these findings with neuron-specific resolution, immunofluorescence and TUNEL staining were performed. When hAECs-Exos were applied at reperfusion onset, HT22 cells exhibited enhanced expression of the neuronal marker MAP2, and BV2 cells displayed significantly reduced apoptosis (*p* < 0.01 and *p* < 0.001, respectively), confirming the direct cytoprotective effects of hAECs-Exos on both neuronal and microglial populations. Importantly, TUNEL signals were analyzed specifically in MAP2^+^ neuronal masks, ensuring that neuronal apoptosis was evaluated independently from other cell types (Figure 3A,B).

Furthermore, ELISA assays revealed that post-ischemic administration of hAECs-Exos significantly mitigated OGD/R-induced elevations in the levels of pro-inflammatory cytokines (IL-1β, IL-6, TNF-α) and MDA (a marker of oxidative stress) (*p* < 0.05, *p* < 0.01, *p* < 0.001), while concurrently restoring the activity of SOD (an antioxidant enzyme) (*p* < 0.05, *p* < 0.01) (Figure 3C–E). Consistently, Western blot analysis further showed that hAECs-Exos upregulated the expression of anti-apoptotic proteins Bcl-2 and BDNF, while downregulating the levels of pro-apoptotic proteins, including Bax and cleaved Caspase-3 (*p* < 0.05, *p* < 0.01, *p* < 0.001) (Figure 3F).

Collectively, these results indicate that hAECs-Exos exert multi-faceted neuroprotective effects by mitigating inflammation, alleviating oxidative stress, and modulating intrinsic apoptotic signaling pathways in OGD/R-injured neuronal and microglial cells.

### 3.3. RNA-Seq Analysis and Validation of hAECs-Exos Effects in OGD/R-Injured HT22 and BV2 Cells

To dissect cell-type-specific transcriptional responses to OGD/R and hAEC-derived exosomes, RNA-seq analyses were conducted independently for HT22 neurons and BV2 microglia. All differential expression, clustering, and enrichment results were derived from separate bioinformatic pipelines. Although results are presented in an integrated figure layout (Figure 4), data interpretation strictly differentiates the two cell types.

In HT22 cells, a total of 5355 DEGs were identified between the OGD/R and control groups (2951 upregulated and 2404 downregulated), while 1907 DEGs (1315 upregulated and 592 downregulated) were found between the hAECs-Exos-treated and OGD/R groups. In BV2 cells, 4585 DEGs were identified between the OGD/R and control groups (2464 upregulated and 2121 downregulated), and 1338 DEGs (1144 upregulated and 194 downregulated) between the hAECs-Exos-treated and OGD/R groups.

Hierarchical clustering heatmaps demonstrated distinct transcriptomic profiles across control, OGD/R, and hAECs-Exos-treated groups in both HT22 and BV2 cells (Figure 4A). KEGG pathway enrichment analysis of DEGs reversed by hAECs-Exos revealed cell-type-specific enrichment patterns: in HT22 neurons, pathways related to PI3K-AKT signaling, neurotrophin signaling, and cell cycle regulation were prominent, whereas BV2 microglia showed enrichment in cytokine–cytokine receptor interaction, p53 signaling, and inflammatory modulation (Figure 4B).

A Venn diagram analysis further identified 22 overlapping genes that were downregulated by OGD/R and subsequently upregulated following hAECs-Exos treatment in both cell types (Figure 4C). These shared DEGs were primarily involved in processes including inflammation resolution, oxidative stress regulation, and tissue remodeling—hallmark mechanisms of exosome-mediated neuroprotection.

Among the 22 overlapping DEGs reversed by hAECs-Exos in both HT22 and BV2 cells, Forkhead box protein M1 (FoxM1) was prioritized for further investigation, based on its pronounced transcriptional responsiveness, established role in stress-response regulation, and consistent correlation with neuroprotective phenotypes observed in our preliminary experiments. FoxM1, a member of the Forkhead box (Fox) transcription factor family, has been extensively studied as a central regulator of cellular processes such as proliferation, apoptosis, and inflammation. While its dysregulation has been implicated in the progression of various human malignancies, including enhanced invasion, migration, and metastasis [21,22,23], the role of FoxM1 in the context of cerebral ischemia, particularly as a potential mediator of hAEC-derived exosome effects, remains poorly understood and warrants focused mechanistic exploration.

Transcriptomic profiling further confirmed the responsiveness of FoxM1 to both injury and treatment: in HT22 neurons, OGD/R significantly downregulated FoxM1 expression, which was remarkably restored following hAECs-Exos treatment; in BV2 microglia, where OGD/R had minimal impact, hAECs-Exos still induced a marked upregulation (Figure 4D). These observations suggest FoxM1 may act as a central transcriptional switch mediating protective reprogramming.

Moreover, FoxM1 was found to be abundantly expressed within hAECs-Exos themselves (Figure 4E), suggesting a possible mechanism of paracrine delivery. RT-qPCR and Western blot validation confirmed that FoxM1 mRNA and protein levels were suppressed by OGD/R and significantly reversed by hAECs-Exos in both HT22 and BV2 cells (*p* < 0.001) (Figure 4F).

Collectively, these findings preliminarily identify FoxM1 as a critical regulatory molecule mediating the effects of hAECs-Exos, and indicate its key role in driving protective transcriptomic reprogramming of neurons and microglia under ischemic stress.

### 3.4. hAECs-Exos Ameliorate OGD/R-Induced Neuronal Injury via FoxM1

Based on transcriptomic and protein-level analyses, FoxM1 was preliminarily identified as a pivotal mediator of the neuroprotective effects exerted by hAECs-Exos. To validate its functional role, FoxM1 knockdown (sh-FoxM1) was induced in HT22 and BV2 cells via lentiviral shRNA transfection. RT-qPCR and Western blot analyses confirmed efficient and specific knockdown of FoxM1 in hAECs and their derived exosomes, including comparison with the sh-FoxM1-NC control group (Figure 5A,B).

Notably, hAECs-Exos with FoxM1 knockdown failed to restore FoxM1 expression in OGD/R-treated HT22 and BV2 cells (Figure 5C,D). Functionally, FoxM1 knockdown abolished the neuroprotective benefits conferred by hAECs-Exos against OGD/R-induced injury. The improvements in cell viability, suppression of inflammatory cytokines, and reduction in oxidative stress (MDA/SOD levels) observed in wild-type cells were no longer evident in FoxM1-deficient models (Figure 5E–H).

Furthermore, the modulation of apoptosis-related proteins by hAECs-Exos—including upregulation of Bcl-2 and BDNF, and downregulation of Bax and cleaved Caspase-3—was lost upon FoxM1 silencing (Figure 5I,J).

These findings strongly support that FoxM1 is an essential downstream effector mediating the anti-apoptotic, anti-inflammatory, and antioxidative actions of hAECs-Exos in ischemic injury models.

## 4. Discussion

Both hAECs and their derived exosomes have demonstrated neuroprotective potential in mitigating nervous system injuries and promoting neural repair [24,25,26]. Across various experimental models, interventions with either hAECs or hAECs-Exos—administered during both acute and delayed phases of ischemia—have consistently been shown to reduce infarct volume, alleviate histopathological brain damage, enhance neuronal survival and regeneration, and inhibit neuroinflammation and apoptosis. Notably, hAECs-Exos offer superior safety and translational potential compared to their parental cells due to their cell-free nature and lower immunogenicity [17,27].

In the present study, we confirmed the efficient internalization of hAECs-Exos by both HT22 neurons and BV2 microglial cells. Exosome treatment, whether applied as a preconditioning agent or during reperfusion, effectively attenuated OGD/R-induced cellular injury by suppressing oxidative stress and inflammatory responses while restoring cell viability. Through RNA sequencing and molecular-level validation, we identified FoxM1 as a key downstream effector molecule regulated by hAECs-Exos, and its expression was significantly restored following exosome treatment after OGD/R-induced injury.

hAECs possess a unique hybrid profile of embryonic and adult stem cell characteristics—combining pluripotency with strong immunomodulatory capacity—and exhibit practical advantages including ethical acceptability, high yield, and low immunogenicity [28]. In regenerative medicine, exosomes are increasingly recognized as potent paracrine effectors of stem cells. Compared to whole-cell therapies, hAECs-Exos circumvent the challenges associated with stem cell transplantation, such as tumorigenicity and poor engraftment. Their nanoscale size and membrane composition facilitate blood–brain barrier (BBB) penetration and targeted biodistribution [19]. In the OGD/R model, we observed significant therapeutic benefit from hAECs-Exos, particularly when administered during the reperfusion phase, highlighting their clinical potential as a delayed-phase intervention for ischemic stroke.

FoxM1 is a transcription factor characterized by its conserved forkhead DNA-binding domain and its central role in cell proliferation, cycle progression, DNA repair, and immune regulation. While FoxM1 has been widely studied as an oncogenic driver in tumor biology and is considered a diagnostic and prognostic biomarker for various cancers [21,29], its functions extend beyond oncology. Emerging evidence suggests that FoxM1 contributes to neurogenesis, neuronal survival, and brain repair processes following injury [30]. In the context of central nervous system disorders, FoxM1 has been shown to mediate neuronal differentiation and protect against neurodegeneration. For instance, in Xenopus laevis embryos, FoxM1 links cell cycle progression with early neuronal fate determination [31]. In rodent models of tMCAO, sustained reperfusion leads to downregulation of nuclear FoxM1, correlating with increased neuronal apoptosis [32]. Moreover, studies have shown that activation of upstream regulators like Wnt3a can upregulate FoxM1 expression and reduce neuronal damage, further supporting its neuroprotective role [33].

FoxM1 stands out as a novel focus in ischemic stroke research because of its ability to integrate multiple pathological pathways, which stands in sharp contrast to other transcription factors previously studied in stroke models. For instance, unlike NF-κB (primarily involved in pro-inflammatory signaling), Nrf2 (which regulates oxidative stress), or HIF-1α (focused on hypoxia adaptation) [34,35,36], FoxM1 functions as a pleiotropic regulatory hub that orchestrates key processes in post-ischemic neuroprotection—including cell cycle progression, anti-apoptosis, DNA repair, mitochondrial function, and neurogenesis. Recent preclinical studies have also highlighted its involvement in CNS repair: for example, Matei et al. demonstrated that intranasal Wnt3a administration upregulated FoxM1 via the Frizzled-1/PIWI1a axis and reduced infarct volume following stroke, while Jia et al. reported that FoxM1 modulates a miRNA network governing neural stem cell homeostasis [33,37]. These findings suggest that FoxM1 may play a broader role in brain injury recovery than previously appreciated and justify its investigation in the context of exosome-mediated therapy.

In our study, OGD/R insult significantly downregulated the mRNA and protein of FoxM1 in both HT22 and BV2 cells. Importantly, treatment with hAECs-Exos restored FoxM1 expression, suggesting that exosomal delivery contributes to FoxM1-mediated neuroprotection. To validate this regulatory relationship, we employed shRNA-mediated knockdown of FoxM1 in hAECs. Functional assays confirmed that the absence of FoxM1 abolished the protective effects of hAECs-Exos, including the suppression of oxidative stress, inflammation, and apoptosis. These findings identify FoxM1 as a pivotal downstream effector and transcriptional hub mediating the beneficial effects of hAECs-Exos in ischemic neuronal injury.

Notably, our study not only validates FoxM1 as a key downstream effector of hAECs-Exos but also positions it as a superior target compared to canonical stroke-related transcription factors, offering a more comprehensive strategy for interrupting the ischemic cascade and improving stroke outcomes. Despite these promising findings and the mechanistic insights gained, it is important to recognize the current study’s limitations and areas that warrant further investigation to enhance translational relevance.

Although our findings provide compelling evidence for the neuroprotective effects of hAECs-Exos and highlight FoxM1 as a key mediator in this process, several limitations must be acknowledged:

First, while we demonstrated efficient uptake of hAECs-Exos by neuronal and microglial cells and confirmed their protective effects in vitro, the current study lacks in vivo validation. The role of FoxM1 in regulating neuroprotection following ischemic stroke remains to be thoroughly investigated in animal models that recapitulate the complex neurovascular environment of the brain.

Second, although RNA-seq and protein-level data support FoxM1 as a central regulatory factor, the upstream signals responsible for exosome-mediated FoxM1 modulation remain unclear. Whether hAECs-Exos deliver transcriptionally active FoxM1 mRNA/protein or indirectly modulate its expression through cargo such as miRNAs or signaling lipids requires further investigation.

Third, although our knockdown experiments demonstrated that FoxM1 is required for hAECs-Exos-mediated protection, the absence of reciprocal rescue (overexpression) experiments limits our ability to fully resolve causality: we cannot rule out that FoxM1 upregulation is correlative rather than directly driving the protective effects. In future studies, we plan to deploy FoxM1 overexpression vectors or CRISPR activation systems to test whether elevating FoxM1 levels alone can recapitulate or enhance the exosome-induced protection. Notably, prior work in a different system (e.g., mesenchymal stem cells overexpressing FoxM1 in an ARDS model) has shown that FoxM1 overexpression can augment tissue repair, offering supportive precedent for such an approach [38].

Fourth, while our knockdown experiments demonstrated that FoxM1 is required for hAECs-Exos-mediated protection, we did not perform reciprocal rescue experiments (e.g., FoxM1 overexpression in the absence of exosome treatment) to formally establish causality. Such experiments would strengthen the conclusion that FoxM1 upregulation is not merely correlated with, but directly responsible for, the neuroprotective effects observed. Future studies will employ FoxM1 overexpression vectors to verify whether forced FoxM1 upregulation can recapitulate the protective phenotype induced by hAECs-Exos, thereby solidifying its causal role.

Lastly, the downstream targets and signaling pathways controlled by FoxM1 in the context of ischemic injury are not fully delineated. Future studies should focus on identifying the transcriptional networks governed by FoxM1, including interactions with oxidative stress response genes, cytokine signaling pathways, and neurotrophic factors. These insights may uncover new therapeutic targets and refine exosome-based treatment strategies.

## 5. Conclusions

By systematically optimizing the OGD/R model and integrating functional, biochemical, and transcriptomic analyses, this study demonstrates that hAEC-derived exosomes (hAECs-Exos) confer significant neuroprotective effects during both acute and delayed phases of ischemic injury. These findings support their potential as a therapeutic modality with a broadened intervention window for ischemic stroke.

Mechanistically, our in vitro results identify FoxM1 as a critical downstream effector involved in hAECs-Exos-mediated protection. Restoration of FoxM1 expression appears essential for reversing OGD/R-induced neuronal damage, including apoptosis, inflammation, and oxidative stress.

While further validation, particularly in vivo studies, is necessary to fully elucidate the role of FoxM1 and its regulatory network in the context of intact biological systems, our findings offer robust in vitro evidence to support the continued exploration of exosome-based therapeutic strategies. This work primarily advances the understanding of the molecular mechanisms governing exosome-mediated neuroprotection and explicitly frames in vivo validation of the hAECs-Exos/FoxM1 axis as a key next step to move toward potential clinical application in stroke treatment.

## Figures and Tables

**Figure 1 biology-14-01368-f001:**
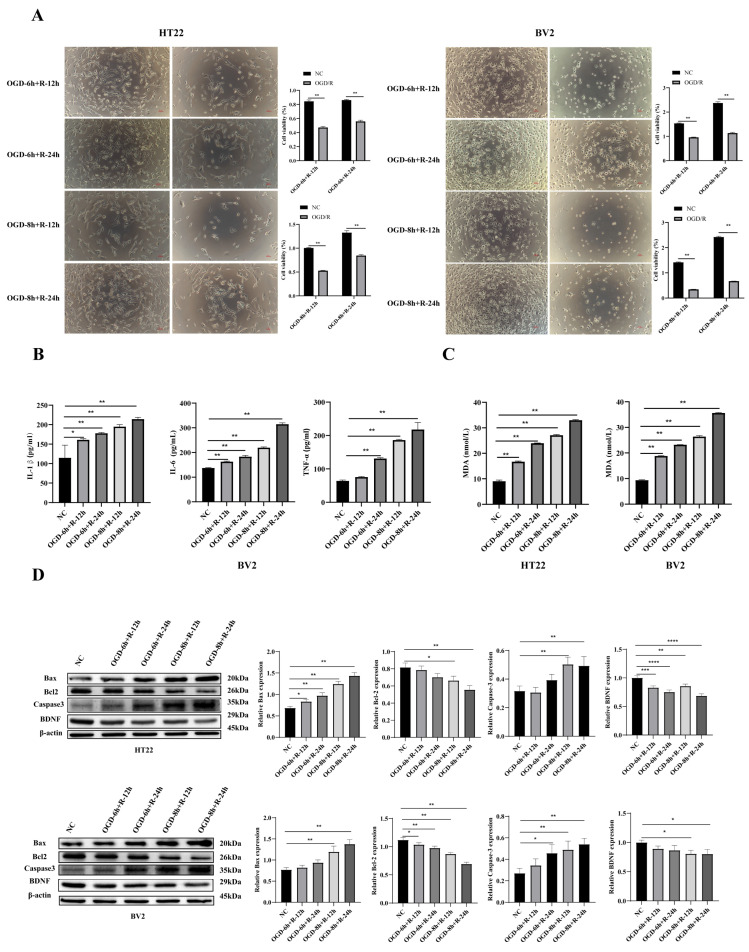
Effects of OGD/R duration on cellular injury in HT22 and BV2 cells. (**A**) CCK-8 assay results and phase-contrast microscopy revealed time-dependent viability loss in both HT22 (**left panel**) and BV2 (**right panel**) cells. (**B**) ELISA quantification showed that OGD/R significantly increased secretion of IL-1β, IL-6, and TNF-α in BV2 cells. (**C**) MDA levels, as an indicator of oxidative stress, were elevated in both cell types after OGD/R. (**D**) Western blot analysis showed increased ex-pression of pro-apoptotic Bax and cleaved caspase-3, and reduced levels of anti-apoptotic Bcl-2 and neurotrophic factor BDNF. Data are presented as mean ± SD from at least three independent experiments. Statistical significance was determined by one-way ANOVA followed by Tukey’s post hoc test. * *p* < 0.05, ** *p* < 0.01, *** *p* < 0.001, **** *p* < 0.0001 vs. NC group.

**Figure 2 biology-14-01368-f002:**
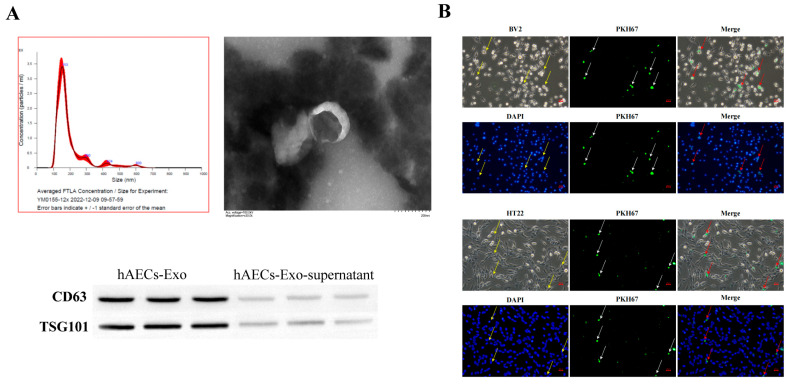
Characterization of hAEC-derived exosomes (hAECs-Exos). (**A**) Exosomes exhibited typical cup-shaped morphology under transmission electron microscopy (TEM), displayed a defined particle size distribution and concentration as assessed by nanoparticle tracking analysis (NTA), and expressed the exosomal surface markers CD63 and TSG101 as confirmed by Western blot. (**B**) Confocal fluorescence imaging demonstrated effective internalization of PKH67-labeled exo-somes (green) by both HT22 and BV2 cells, with DAPI staining (blue) used to visualize cell nuclei. Yellow and red arrows indicate the intracellular localization of PKH67-labeled exosomes.

**Figure 3 biology-14-01368-f003:**
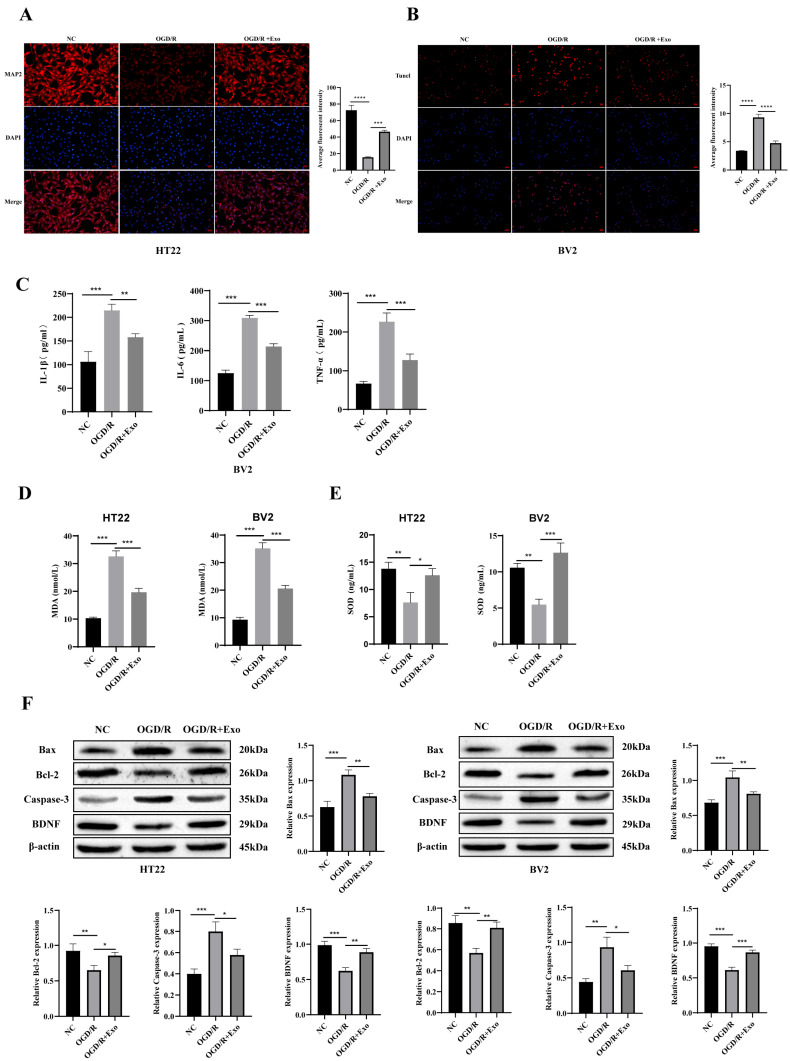
Functional evaluation of hAEC-derived exosomes (hAECs-Exos) on HT22 and BV2 cells subjected to OGD/R. (**A**) Immunofluorescence staining showed that hAECs-Exos restored the expression of MAP2, a neuron-specific marker, in HT22 cells (MAP2, red; nuclei, blue). (**B**) TUNEL staining demonstrated that hAECs-Exos reduced apoptosis in BV2 cells (TUNEL-positive nuclei, red; DAPI-stained nuclei, blue). (**C**–**E**) ELISA results indicated that hAECs-Exos suppressed OGD/R-induced secretion of pro-inflammatory cytokines (IL-1β, IL-6, and TNF-α) in BV2 cells, reduced levels of the oxidative stress marker malondialdehyde (MDA), and restored superoxide dismutase (SOD) activity in both HT22 and BV2 cells. (**F**) Western blot analysis showed reversal of OGD/R-induced apoptotic signaling, with increased levels of the anti-apoptotic protein Bcl-2 and brain-derived neurotrophic factor (BDNF), and decreased levels of pro-apoptotic Bax and cleaved caspase-3. Data are presented as mean ± SD from at least three independent experiments. Statistical significance was determined using one-way ANOVA followed by Tukey’s post hoc test. * *p* < 0.05, ** *p* < 0.01, *** *p* < 0.001, **** *p* < 0.0001 vs. NC or OGD/R group.

**Figure 4 biology-14-01368-f004:**
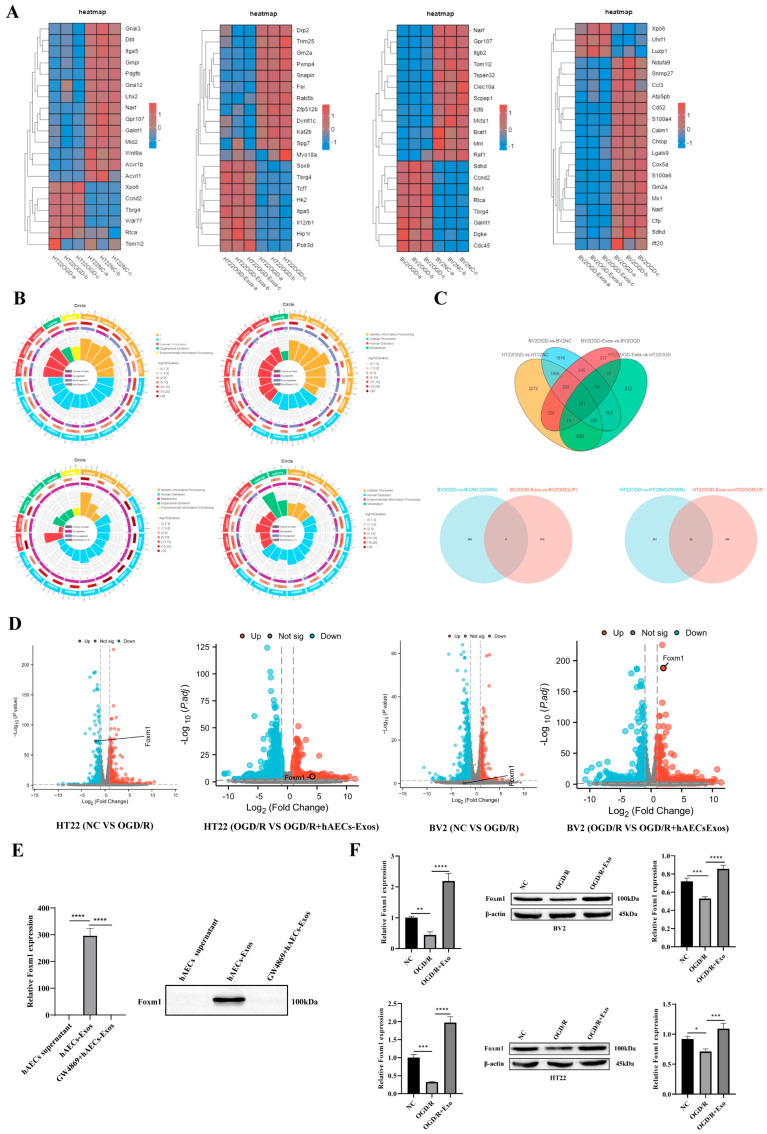
Transcriptomic profiling and validation of FoxM1 regulation in HT22 and BV2 cells following OGD/R and hAEC-derived exosome (hAECs-Exos) treatment. (**A**) Hierarchical clustering heatmap of representative differentially expressed genes (DEGs) across control, OGD/R, and hAECs-Exos-treated groups in HT22 and BV2 cells. (**B**) Kyoto Encyclopedia of Genes and Genomes (KEGG) pathway enrichment presented as a chord diagram, highlighting pathways significantly modulated by hAECs-Exos, including p53 signaling, PI3K-AKT signaling, cell cycle regulation, and focal adhesion. (**C**) Venn diagram showing 22 overlapping DEGs that were downregulated by OGD/R and upregulated upon hAECs-Exos treatment, primarily associated with inflammation, oxidative stress, and cellular repair. (**D**) Volcano plot illustrating significant upregulation of Foxm1 expression in hAECs-Exos-treated groups relative to OGD/R-only groups. (**E**) Western blot validation of FoxM1 expression and the effect of GW4869, an inhibitor of exosome secretion, on exosome-mediated signaling. (**F**) RT-qPCR and Western blot analyses of FoxM1 levels in HT22 and BV2 cells showed that OGD/R suppressed FoxM1 expression, which was significantly restored following hAECs-Exos treatment. Data are presented as mean ± SD from at least three independent experiments. Statistical significance was determined using one-way ANOVA followed by Tukey’s post hoc test. * *p* < 0.05, ** *p* < 0.01, *** *p* < 0.001, **** *p* < 0.0001 vs. NC or OGD/R group.

**Figure 5 biology-14-01368-f005:**
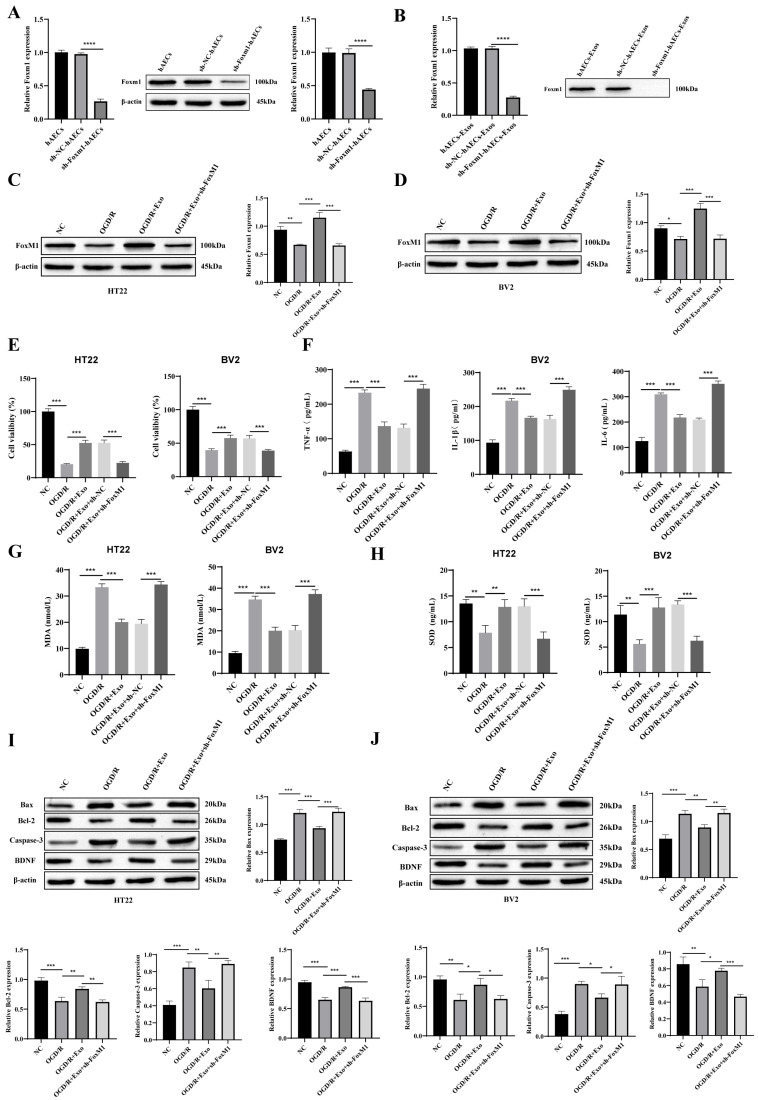
FoxM1 knockdown abolishes the neuroprotective effects of hAEC-derived exosomes (hAECs-Exos) in OGD/R-treated HT22 and BV2 cells. (**A**,**B**) RT-qPCR and Western blot analyses confirmed efficient and specific knockdown of FoxM1 in hAECs and their derived exosomes. The inclusion of the sh-FoxM1-NC group verified knockdown specificity. (**C**,**D**) Western blot showed that FoxM1 knockdown in hAECs-Exos abolished their ability to restore FoxM1 expression in OGD/R-treated HT22 and BV2 cells. (**E**) CCK-8 assays indicated that the cytoprotective effect of hAECs-Exos was eliminated upon FoxM1 knockdown. (**F**) ELISA demonstrated that the anti-inflammatory effects of hAECs-Exos, including reduced secretion of TNF-α, IL-1β, and IL-6, were significantly weakened in FoxM1-deficient conditions. (**G**,**H**) FoxM1 knockdown attenuated the antioxidative effects of hAECs-Exos, as evidenced by elevated malondialdehyde (MDA) levels and decreased superoxide dismutase (SOD) activity in both cell types. (**I**,**J**) Western blot analyses showed that hAECs-Exos failed to reverse OGD/R-induced alterations in apoptosis-related proteins—including increased Bax and cleaved Caspase-3 and decreased Bcl-2 and brain-derived neurotrophic factor (BDNF)—in the absence of FoxM1. Data are presented as mean ± SD from at least three independent experiments. Statistical significance was determined using one-way ANOVA followed by Tukey’s post hoc test. * *p* < 0.05, ** *p* < 0.01, *** *p* < 0.001, **** *p* < 0.0001 vs. NC, OGD/R, or OGD/R + Exo groups.

**Table 1 biology-14-01368-t001:** Primer sequences for target genes in qPCR.

Name	Forward Primer (5′-3′)	Reverse Primer (5′-3′)
*h-FoxM1*	*GAGACCTGTGATGGTGAGGC*	*ACCTTAACCTGTCGCTGCTC*
*m-FoxM1*	*AAGCCTGAGCCAAGTATCTCG*	*GCCTCTTCAACCTGTTGCC*
*h-GAPDH*	*CTGGGCTACACTGAGCACC*	*AAGTGGTCGTTGAGGGCAATG*
*m-GAPDH*	*AGGTCGGTGTGAACGGATTTG*	*TGTAGACCATGTAGTTGAGGTCA*

## Data Availability

All data supporting the findings of this study are available from the corresponding author upon reasonable request.

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
