# Peer review of "Transcriptomic Investigation of FoxM1-Mediated Neuroprotection by hAEC-Derived Exosomes in an In Vitro Ischemic Stroke Model"

_biology, 2025, doi:10.3390/biology14101368_

Round 1

Reviewer 1 Report

Comments and Suggestions for Authors

Transcriptomic Investigation of FoxM1-Mediated Neuroprotection by hAEC-Derived Exosomes in an In Vitro Ischemic Stroke

In this study, the authors explores the exosomes and FoxM1 protective effects in Cell Culture of Mouse hippocampal neuronal HT22 cells and murine microglial BV2 cells .

However, in the method, it is good to add and clarify the following information:

The Method,

Cell Culture and OGD/R Model Construction

Neuron–microglia crosstalk can dominate outcomes (e.g., microglial cytokines potentiating neuronal death). Separate cultures measure cell-intrinsic responses; Interpretation of mechanisms could be affected.

The neuron-specific viability is the right move, and an antibody-based assay is a solid choice. CCK-8 is a bulk metabolic readout; it can’t tell you how many HT22 (neurons) survived—especially if there’s any co-culture with BV2. An immunostaining assay lets you count neurons specifically and score their death/apoptosis with or without exosomes under OGD/R.

Here’s a practical add:

Neuron-specific antibody assay (works for mono- or co-culture)

  1. Treatments: Normoxia, OGD/R, OGD/R+hAEC-Exos (pre- and post-).
  2. Fix at the viability timepoint (e.g., end of reoxygenation) with 4% PFA.
  3. Stain neurons: monoclonal MAP2 or βIII-tubulin (Tuj1) to identify HT22.
  4. Stain apoptosis/viability: monoclonal cleaved caspase-3 or TUNEL; include DAPI for nuclei.

     Cell-type specificity: Only MAP2⁺ cells are counted as neurons.

 Direct survival metric: You get “neurons per field” and “% apoptotic neurons,” not just metabolic activity.

  Co-culture compatible: You can analyze neurons and microglia separately in the same well (MAP2 vs Iba1 masks).

Given that CCK-8 is a bulk assay, a neuron-specific viability readout is needed to support the claim that hAEC-Exos protect neurons under OGD/R. Please add an antibody-based assay (e.g., MAP2 or βIII-tubulin to identify HT22, combined with cleaved caspase-3 or TUNEL to quantify neuronal death) and analyze outcomes on the neuronal mask only. This will disambiguate neuronal protection from microglial effects and metabolic shifts.

Additionally, RNA-seq (bulk) is presented as evidence of neuronal protection. However, bulk RNA-seq does not provide cell-type–specific viability and, with standard normalization, cannot infer cell numbers per well. Please clarify whether sequencing was performed on separate monocultures (HT22-only, BV2-only) or mixed co-cultures. If mixed, we recommend adding a neuron-specific viability assay (e.g., labeled-cell live/dead imaging, MAP2/NeuN counts, Transwell with bottom-well CCK-8, or Annexin V/PI with cell-type gating). RNA-seq should be used to support mechanism rather than substitute for neuronal survival quantification. If you intend to infer abundance from RNA, include spike-ins and a validated deconvolution approach.

However, finally they present the convincing evidence for separate culture result for neurons and microglia found in following figures:

“Figure 3. Transcriptomic profiling and validation of FoxM1 regulation in HT22 and BV2 cells fol- 338 lowing OGD/R and hAECs-Exos treatment. (A) Hierarchical clustering heatmap of representative 339 differentially expressed genes”

And also supportive result sin supplementary:

“Supplementary Materials: Figure S1: Effects of hAECs-derived exosome treatment at different time 476 points on cell viability of HT22 and BV2 cells following OGD/R injury. Representative morphologi- 477 cal images and CCK-8 quantification showing”

Reviewer 2 Report

Comments and Suggestions for Authors

Thank you for the opportunity to review this interesting manuscript. The work is timely and well written and the idea of using hAEC-derived exosomes in ischemic stroke is a interesting approach. The experiments are well done overall, but there are several points to consider that could benefit your hard work and need attention before the paper is ready for publication. 

  1. The introduction overall looks good but consider emphasize more clearly why FoxM1 is a novel focus compared to other transcription factors already studied in stroke models. This will help highlight the originality. Additionally the transitions between paragraphs could be improved.
  2. For the exosome characterization, the methods describe TEM, NTA, and marker validation, but please add more details on particle-to-protein ratio or yield, and also clarify how you avoided contamination from FBS-derived vesicles during culture.
  3. RNA-seq data section, the workflow is well explained, but I could not find sequencing mQC information (my apologies in advance if I missed this). PElase provide detailas as read depth, alignment %, RIN values, and ideally deposit the complete DEG lists in the supplemental material for transparency.
  4. Regarding the FoxM1 experiments the shRNA is convincing, but no rescue experiment (overexpression of FoxM1) is included. This should at least be mentioned as a limitation since it leaves some uncertainty about causality.
  5. As for the figures: Please include exact p-values where possible instead of only thresholds. Supplementary Figure S1 (pre- vs post -treatment is relevant and should be emphasized more in the results. I noticed  what it looks like a possible duplication in the Caspase-3 Western blot in figure 1. i suggest carefully reviewing this point and provide clarification or corrected data if needed.
  6. Discussion/conclusion claim of clinical translatability should be moderate, as findings are currently limited to in vitro models. I strongly recommend  to focus on mechanistic insights and framing in vivo validation as the next step. 
Comments on the Quality of English Language

The English is generally clear but could benefit from minor polishing to rephrase and reduce long sentences that could be streamlined for clarity and enhance readability. There are some slight grammar details that could be improved.

Reviewer 3 Report

Comments and Suggestions for Authors

Comments for authors:

  1. Authors have mentioned that aim of the study is to elucidate regulatory pathways mediated by hAECs exosome. But authors have selected one molecule and have shown presence or absence of the molecule and its effect on cell viability, certain proinflammatory cytokines and MDA as oxidative stress marker. I appreciate the design of study but it does not shed light on regulatory pathways. So, maybe authors can rewrite this sentence!

2.Meaterials and methods- well written. Easy to understand. Please mention passage number of cell culture for experiments.

  1. Fig2- PanelC- this assay is for which cell line? Also, if authors suggest that exosomes are supplementing the HT22 and BV2 with FOXM1 molecule, they should show presence of FOXM1 in exosome by WB, like they have valiated CD63 and TSG101. This will increase the impact of the study.

4.Fig2- there is no NC+EXO group. Addition of this group will serve as control for exosome application experiment.

  1. Fig3-Venn diagram is not visible and legible. Adding FOXM1 mRNA expression will validate RNAseq data. Validating atleast 1 target of FOXM1 will suggest that molecule is directly functionally involved in protective effect.
  2. Fig4- There should be scrambled sh-FOXM1-NC group to validate how much knock down has happened for the molecule. I am sure I have not missed it, but I don’t see any data suggesting % knockdown of the molecule in both cell types in first place. Please add this data so that rest of the data make more sense.
  3. There should be more background about why only FOXM1 was chosen for further study
